# Coupled Segmentation and Edge Learning via Dynamic Graph Propagation

**Zhiding Yu**[*]  **Rui Huang**[*†], **Wonmin Byeon, Sifei Liu, Guilin Liu,**
**Thomas Breuel, Anima Anandkumar, Jan Kautz**

NVIDIA

## Abstract

Image segmentation and edge detection are both central problems in perceptual grouping. It is therefore interesting to study how these two tasks can be coupled to benefit each other. Indeed, segmentation can be easily transformed into contour edges to guide edge learning. However, the converse is nontrivial since general edges may not always form closed contours. In this paper, we propose a principled end-to-end framework for coupled edge and segmentation learning, where edges are leveraged as pairwise similarity cues to guide segmentation. At the core of our framework is a recurrent module termed as dynamic graph propagation (DGP) layer that performs message passing on dynamically constructed graphs. The layer uses learned gating to dynamically select neighbors for message passing using max-pooling. The output from message passing is further gated with an edge signal to refine segmentation. Experiments demonstrate that the proposed framework is able to let both tasks mutually improve each other. On Cityscapes validation, our best model achieves 83.7% mIoU in semantic segmentation and 78.7% maximum F-score in semantic edge detection. Our method also leads to improved zero-shot robustness on Cityscapes with natural corruptions (Cityscapes-C).

## 1  Introduction

Image segmentation and edge detection have been widely studied as important perception problems. The two problems are closely related. In fact, segmentation subsumes edge detection since any segmentation contour makes a closed boundary of a region. The converse is however not true since general edges do not always form closed contours. Nevertheless, edge detection can serve as an auxiliary task to improve segmentation performance since edges provide important pairwise similarity cues for segmentation. Early works tend to focus on the grouping and contrast of pixels from a perceptual similarity perspective. Martin et al. [1] proposed the Berkeley Segmentation Dataset, a popular benchmark for segmentation and boundary detection that inspired many impactful works in perceptual grouping [2–5]. The recent surge of deep learning renders powerful representations with learned features using convolutional neural networks (CNNs) [6]. This has led to great advances in both areas [7–12], but the two tasks are often considered separately.

In light of the status quo, we consider coupled edge and segmentation learning. Our goal is two-fold: (1) Multi-task learning - being able to produce high quality edge detection and segmentation. (2) Mutual improvement - the two tasks can help each other with non-trivial performance gains. Designing a principled framework is however nontrivial. The key question is how sparse edge signals can be effectively transformed into dense region-level ones to interact with segmentation. To this end, we propose a learnable recurrent message passing layer where semantic edges are considered

---

[*]Equal contribution. Correspondence to Zhiding Yu <zhidingy@nvidia.com>.
[†]Work partially done during an internship at NVIDIA.

35th Conference on Neural Information Processing Systems (NeurIPS 2021).

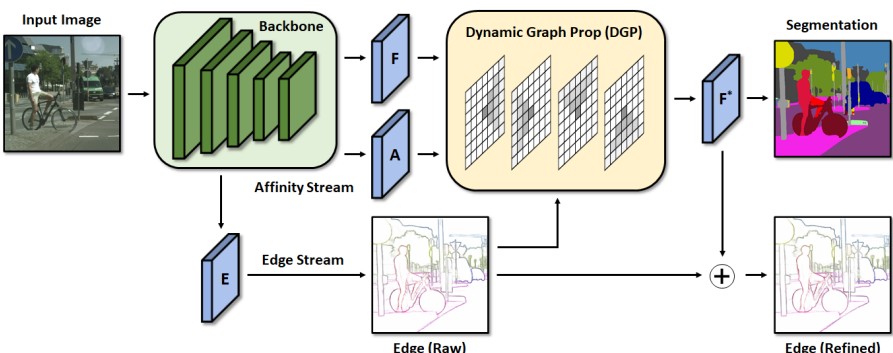

Figure 1: **Framework Overview.** The backbone network provides the encoded features as well as an edge stream that produces a semantic edge map. The DGP layer takes them as inputs and uses learnable message passing to produce a refined feature map (F*) to predict segmentation and edges.

as explicitly learned gating signals to refine segmentation. An overview of our framework is shown in Fig. 1. Specifically, the dynamic message passing layer uses affinity gates to select the neighbor for message passing using max-pooling. It conducts message passing sweeps in each of the four directions: left to right, right to left, top to bottom and bottom to top. The message passing is jointly gated by both the affinity and edge gates, therefore allowing edge cues to naturally influence long-range dense predictions. As such, our framework presents a context module that is clean, compact yet powerful. Our technical contributions can be summarized as follows:

- We formulate recurrent message passing as dynamic graph propagation (DGP). We show that such a formulation simplifies the required normalization in propagation networks [13]. It also dynamically finds graph structures that encodes pixel affinities and improves dense prediction.

- We propose a double-gate design where message passing is jointly gated by both the affinity and edge gates to refine the segmentation feature map. We show that this design together with the dynamic 1-way connection in DGP better couples segmentation and edge learning.

- We obtain state-of-the-art results on joint semantic segmentation and edge detection. We also show that DGP leads to strong zero-shot robustness to natural corruptions with significant improvement over prior methods on Corrupted Cityscapes (Cityscapes-C).

Multitasking segmentation and edge learning is desirable for several reasons: 1) There are many downstream applications where both are needed, such as occlusion reasoning [14], localization [15], proposal generation [16–19] and conditional generation [20]. 2) There are many challenging cases where segmentation quality is poor but edge quality is far more superior, e.g. segmentation tends to be inferior near object boundaries since they are often optimized for IoU rather than precision [21]. In these cases, edge learning can potentially capture details missed by segmentation. 3) Implicitly improved model generalization as a result of the coupled learning [22].

## 2   Related Work

**Semantic Segmentation.** There is a rich set of prior work in semantic segmentation. Long et al., [7] proposed fully convolutional end-to-end training of semantic segmentation and pointed out its connection to recognition [6]. Chen et al. [8] introduced atrous convolution and atrous spatial pyramid pooling (ASPP) to capture multi-scale image contexts. Another contemporary work is the wide ResNet-38 which explores a relatively shallower but wider backbone [23]. It has also been shown that context plays an important role in segmentation, including context encoding [24, 25], multi-scale context [26–28] as well as relational context [29, 30]. More recently, there has been a surge of interests in segmentation with Transformers [31–34]

**Boundary/Edge Detection.** Similar to segmentation, boundary/edge detection have been widely studied as perceptual grouping problems in early literature [1–3]. Recent methods tend to resort to CNNs. For example, Bertasius et al. proposed a multi-scale deep network architecture for top-down contour detection, where as Xie et al. [11] further introduced holistically-nested edge detection

(HED) for end-to-end edge learning. Besides detecting binary edges, Hariharan et al. [35] proposed the Semantic Boundaries Dataset (SBD) which has become a popular benchmark for semantic edge detection. Compared to binary edge detection, semantic edge detection involves the semantic classification of edge pixels in addition to localization which presents additional challenges to existing frameworks. A series of works including HFL [36], CASENet [12], SEAL [37] and STEAL [38] have followed up [35] and pushed the boundaries.

**Multi-task Segmentation and Edge Learning.** Multi-task segmentation and edge learning remains under-studied but is not entirely new. Edges have been pairwise similarity cues to improve segmentation [3, 39, 18] and superpixels [4]. It was shown that edges can be transformed into dense regions through the Laplacian Eigenmaps of boundary transformed affinity [36, 5, 40]. Despite being robust, these methods are generally slow and not end-to-end trainable. For more recent CNN based models, end-to-end multi-task learning on a shared backbone is a natural choice [41]. In addition, proper regularization between segmentation and edge has been shown to improve the performance of both tasks. For example, Takikawa et al. [21] use softmax with temperature to impose consistency between segmentation and semantic edges. Zhen et al. [42] correlate semantic segmentation and edge detection tasks with a consistency loss. Our work goes beyond multi-tasking and let the two task more deeply coupled through dynamic graph propagation.

**Structured Dense Prediction.** Segmentation is a dense prediction task where structured information can be useful. Structured prediction models such as Markov random fields (MRFs) [43], conditional random fields (CRFs) [44–46] and energy minimization [47, 48] have widely proved helpful in segmentation problems by imposing contrast-sensitive smoothness. In addition, structured inference can also be unfolded as network layers for end-to-end training [49, 50], therefore combining the advantages from both ends. Recently, there is an increasing trend to directly model the message passing process itself using multi-dimensional RNN [51–53], graph RNN [54, 55] and spatial propagation network (SPN) [13, 56, 57]. A common advantage is that these methods render more context-aware prediction with larger receptive fields while preserving local details similar to CRFs. Their ability to train end-to-end allows more powerful representation of inter-pixel similarities and thus better dense prediction quality. Our work can be broadly categorized into this category. It is worth mentioning that the contrast sensitive smoothness term in CRF and propagation networks [52, 13] is an implicit modeling of edge signals by learning to relax smoothness constraints at high contrast areas. There have also been variants of SPN [58] that take binary edge as gate regularization. However, none of these works have explicitly addressed multi-tasking learning with category-aware edges whereas the proposed DGP framework presents a novel and effective solution to this problem.

# 3 Multi-task network

As a first step towards coupled segmentation and edge learning, we introduce two novel multi-task networks that are able to perform multi-task learning for both segmentation and edge detection. An overview of their architectures is shown in Fig. 2. The rest of the paper follows these notations and covers more details for each module.

**Backbone.** A CNN giving a semantic feature map (denoted as $F$) encoding the segmentation information. There could be multiple choices of the backbone networks, ranging from the popular architectures of DeepLabv2 [8], DeepLabv3 [27] to latest state-of-the-arts such as DeepLabv3+ [28] and Dual-Attention Network [30]. Adopting powerful backbones will surely benefit the system-level performance, but the major purpose of this work does not completely lie in achieving state-of-the-arts. We are more interested in showing the effectiveness of a proposed framework on standard backbones. To this end, we consider both CASENet [12] and ResNet-38 [23] as the standard backbones for benchmarking. Both of them comprehensively covers the feature map resolutions of $1$, $1/2$, $1/4$ and $1/8$, with the last resolution being a rule of thumb adopted by many segmentation networks. For both networks, we adopt atrous spatial pyramid pooling layers with dilation rate $[6, 12, 18, 24, 30, 36]$ to capture context information. We set the output channels to be 128 to produce a semantic feature map, fowllowed by either direct segmentation prediction (baselines) or the proposed DGP layer.

**Affinity stream.** Convolution layers that aggregate the cross-layer information of the backbone and produces an affinity feature map (denoted as $A$) to encode the affinity of pairwise neighboring pixels. This stream is only used with the presence of recurrent message passing layer (including its related baselines). The stream aims to model the simialrity of pair-wise neighboring pixels and serve as a

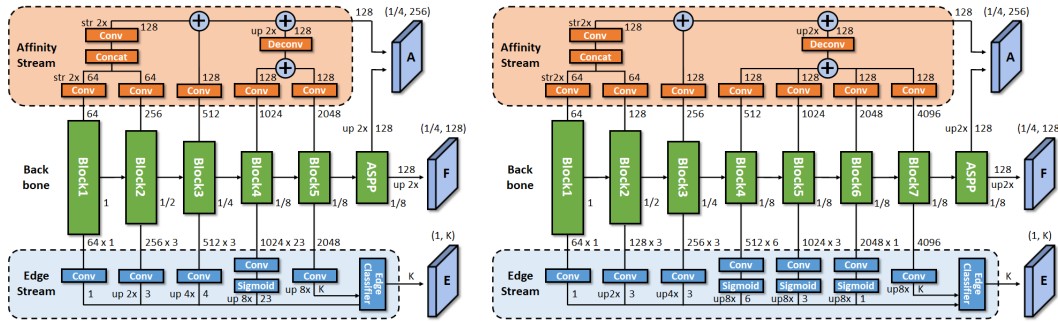

Figure 2: An overview of the architecture for the proposed multi-task Network. Left: architecture with the CASENet backbone. Right: architecture with the ResNet-38 backbone. Each residual block of the backbone is marked with corresponding output scales $(1, 1/2, 1/4, 1/8)$. In addition, the output feature maps $A$, $F$ and $E$ are marked with (scale, channel number).

major input to the gating signal in recurrent message passing layer. Compared to edge stream, the affinity stream seeks to capture coarser-level inter-pixel affinity with slightly deeper convolutions and higher dimensions. The resulting feature map $A$ is a 256 dimensional tensor concatenated by ASPP and the side convolutional feature maps from the affinity stream.

**Edge stream.** Dense skip-layer features with abundant details that are combined with edge classification layer through shared concatenation [12] to produce a semantic edge map (denoted as $E$). We design an improved edge stream over CASENet [12] to better leverage detail information at multiple scales/levels with dense side connections. Unlike CASENet where the bottom blocks only provide 1-channel side features, we densely obtain side features from every sub-block residual modules. In CASENet, this gives side features with a total number of 31 dimensions. Similar rules apply to ResNet-38, where the side feature has a total number of 17 dimensions We also found that applying sigmoid after side features with resolution $1/8$ greatly benefits edge training from two aspects: (1) It helps to stabilize and prevent gradient explosions from dense skip connections. (2) It removes the typical aliasing edge issue caused by upsampling/deconvolution, and produces elegant smooth edges. Even though we do not explicitly apply techniques such as edge alignment [37] in this work, the proposed backbone is able to produce high quality edge predictions under noisy labels. We also notice that it is better to remove sigmoid for side features with higher resolutions. Edge prediction is multi-tasked alongside with ASPP using $1 \times 1$ convolution on top of Res Block5. This returns the $K$ classes coarse edge predictions. Similar to CASENet, we upsample all side features together with the $K$-dimensional edge predictions to full image resolution and apply shared concatenation, where side features are repetitively shared and concatenated with each class for $K$ times, followed by a $K$-way $3 \times 3$ group convolution to produce semantic edges.

**Dynamic graph propagation.** Learnable recurrent message passing layer that takes the above three branches as input to produce a refined semantic feature map (denoted as $F^*$). More details regarding this module will be introduced in the next section.

## 4   Coupled segmentation and edge learning

Although the multi-task network is able to jointly produce segmentation and edge prediction, we are interested in letting these two tasks better coupled to mutually improve each other. To this end, we look into recent spatial propagation networks where edges can be transformed into gating signals that produce long range influence to segmentation. We first give the notations and definitions:

### 4.1   Notations and Settings

We start with a two-dimensional recurrent network architecture with a linear propagation module passing messages (memories) spatially over a feature map. We define a 4-way message passing: 1. Left→ Right (Mode L), Right→Left (Mode R), Top→Bottom (Mode T) and Bottom→Top (Mode B). Each way of message passing will separately generate a hidden state map $H$ which can be approximately viewed as a refined (smoothed) version of $F$. Finally, we take an element-wise max

operation to ensemble them where the model automatically selects the optimal direction with highest neural activation for each pixel:

$$\boldsymbol{F}^* \leftarrow \boldsymbol{H} \triangleq \max\left(\boldsymbol{H}^{\boldsymbol{L}}, \boldsymbol{H}^{\boldsymbol{R}}, \boldsymbol{H}^{\boldsymbol{T}}, \boldsymbol{H}^{\boldsymbol{B}}\right) \tag{1}$$

To perform message passing in each way, one often needs to pre-define a graph that encodes the message passing paths. For tractability and and computation issues, such graph is often sparsely defined with locally connections between immediate neighboring pixels. We follow spatial propagation network (SPN) [13] by defining a three-way local connection, as illustrated on the hand side of Fig. 3. The advantage of such design is obvious: Taking Mode R propagation as an example, one just needs to initiate the message passing from the right most column, and recurrently pass the message from right to left column by column. In this case, the hidden state of each pixel is directly influenced by the three immediate neighboring pixels on the right column. Details of the propagation under Mode R is illustrated on the left hand side of Fig. 3.

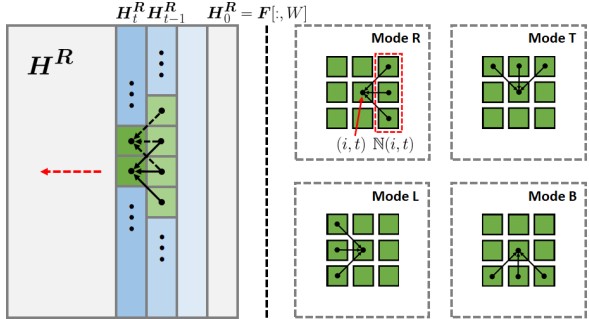

Figure 3: Illustration of spatial propagation. Left: propagation under Right→Left mode. Right: the defined graphs and neighbors of different propagation modes.

Let $\boldsymbol{F} \in \mathbb{R}^{H \times W \times C}$ be the feature map input to the propagation module, $\boldsymbol{H} \in \mathbb{R}^{H \times W \times C}$ the propagation latent space on top of $\boldsymbol{F}$. In addition, let $\boldsymbol{h}_{i,t}$ and $\boldsymbol{f}_{i,t}$ denote the hidden state and feature of the $i$ th pixel located at recurrence $t$[3] on $\boldsymbol{H}$ and $\boldsymbol{F}$, respectively. We denote $\{\boldsymbol{p}_{i,t}^k | k \in \mathbb{N}(i,t)\}$ the set of learnable propagation gating weights for hidden state $\boldsymbol{h}(i,t)$ where $\mathbb{N}(i,t)$ is the set of neighbors of pixel $(i,t)$. In this case, the spatial propagation in each mode is defined as:

$$\boldsymbol{h}_{i,t} = \left(1 - \sum_{k \in \mathbb{N}(i,t)} \boldsymbol{p}_{i,t}^k\right) \odot \boldsymbol{f}_{i,t} + \sum_{k \in \mathbb{N}(i,t)} \boldsymbol{p}_{i,t}^k \odot \boldsymbol{h}_{k,t-1} \tag{2}$$

where $\odot$ is element-wise product, and $\boldsymbol{h}_{k,t-1}$ is the hidden state of pixel $(i,t)$'s neighbors from the previous recurrence. $\{\boldsymbol{p}_{i,t}^k | k \in \mathbb{N}(i,t)\}$ is expandable to an affinity matrix, revealing the global and dense message passing among all the pixels of $\boldsymbol{F}$.

## 4.2 Dynamic graph propagation (DGP)

As a linear module, the above operation requires careful normalization of among $\{\boldsymbol{p}_{i,t}^k | k \in \mathbb{N}(i,t)\}$. In particular, $\sum_{k \in \mathbb{N}(i,t)} \boldsymbol{p}_{i,t}^k \leq 1$ should be satisfied since the energy of the hidden state signal gets unbounded easily under the recurrence operation. To normalize the gating weights, one may consider a linear self-normalization scheme [13] where the constraint $\sum_{k \in \mathbb{N}(i,t)} |p_{i,t}^k| \leq 1$ is imposed to guarantee the stableness of the propagation[4] via dividing each $p_{i,t}^k$ with $\sum_{k \in \mathbb{N}(i,t)} |p_{i,t}^k|$.

But the above formulation also leads to certain limitations. For example, the linear form allows $p_{i,t}^k$ to be both positive and negative, which potentially encourages $p_{i,t}^k$ to be large towards either positive or negative side rather than being monotonic. In addition, when $p_{i,t}^k \geq 0$, $\sum_{k \in \mathbb{N}(i,t)} \boldsymbol{p}_{i,t}^k = 1$ which means that such formulation considers zero unary input from $\boldsymbol{f}_{i,t}$. Therefore, another choice is to constrain $p_{i,t}^k$ with a probabilistic output:

$$\boldsymbol{p}_{i,t}^k = \frac{\exp(\hat{\boldsymbol{p}}_{i,t}^k)}{\sum_{k \in \mathbb{N}(i,t)} \exp(\hat{\boldsymbol{p}}_{i,t}^k)} \odot \sigma(\hat{\boldsymbol{p}}_{i,t}^k), \tag{3}$$

where $\hat{\boldsymbol{p}}_{i,t}^k$ is defined as:

$$\hat{\boldsymbol{p}}_{i,t}^k = \boldsymbol{W}_{\boldsymbol{a}}^\top [\boldsymbol{A}_{i,t}; \boldsymbol{A}_{k,t-1}], \tag{4}$$

---

[3]There is a one-to-one correspondence between $(i,t)$ and pixel index $(h,w)$. But the mapping varies subject to the propagation mode. In Mode R for example, $t$ corresponds to column with $w = W - t$ whereas $h = i + 1$.

[4]Similar property holds when the same constraint is applied to each dimension of $\boldsymbol{p}_{i,t}^k$ in Equation (2).

and $\sigma(\cdot)$ is the Sigmoid function. Note that $\hat{\boldsymbol{p}}_{i,t}^k$ is the raw gating activation without normalization, and is obtained via a linear projection on the concatenated affinity stream features.

The above formation satisfies both $\sum_{k\in\mathbb{N}(i,t)} \boldsymbol{p}_{i,t}^k \leq 1$ and $\boldsymbol{p}_{i,t}^k \geq 0$, while partially considering the unary input with the Sigmoid term. Yet the framework is a bit complicated with many non-linear terms. To this end, we take one more step to further sparsifying and simplifying the gating response by considering a softmax-with-temperature formulation and taking the limit of $T \to 0$:

$$\boldsymbol{p}_{i,t}^k = \lim_{T\to 0} \frac{\exp(\hat{\boldsymbol{p}}_{i,t}^k)/T}{\sum_{k\in\mathbb{N}(i,t)} \exp(\hat{\boldsymbol{p}}_{i,t}^k)/T} \odot \sigma(\hat{\boldsymbol{p}}_{i,t}^k) \tag{5}$$

Substituting Equation (5) into Equation (2), we have:

$$\boldsymbol{h}_{i,t} = \Big(1 - \sigma(\boldsymbol{p}_{i,t}^*)\Big) \odot \boldsymbol{f}_{i,t} + \sigma(\boldsymbol{p}_{i,t}^*) \odot \boldsymbol{h}_{i,t}^*, \tag{6}$$

where the $n$-th dimension of $\boldsymbol{p}_{i,t}^*$ and $\boldsymbol{h}_{i,t}^*$ are defined as:

$$k^* \triangleq \arg\max_{k\in\mathbb{N}(i,t)}\{\hat{\boldsymbol{p}}_{i,t}^k[n]\} \quad \boldsymbol{p}_{i,t}^*[n] = \hat{\boldsymbol{p}}_{i,t}^{k^*}[n] \quad \boldsymbol{h}_{i,t}^*[n] = \boldsymbol{h}_{k^*,t-1}[n]. \tag{7}$$

Equation (6) essentially leads to a dynamic graph propagation (DGP) framework where one performs message passing on a dynamic graph structure by picking neighbor with the highest response. Intuitively, DGP captures a compact structure that has close relation to directed minimum spanning tree, Chu–Liu/Edmonds' algorithm [59] and the recently pro-

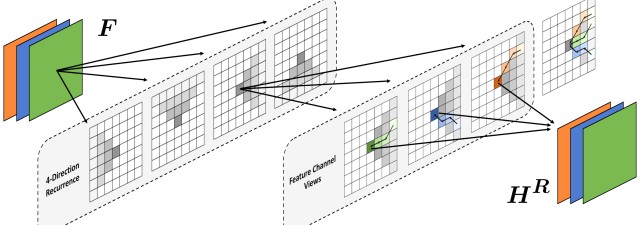

Figure 4: Illustration of dynamic graph propagation.

posed tree filters [60, 61]. Such structure presents an inductive bias that benefits segmentation by filtering out noisy prediction and following only strong signals. Figure 4 illustrates DGP where we show three example paths on three feature channels. Each channel independently takes different paths depending on its own neighbor affinities.

## 4.3 Coupling edge prediction with segmentation

To deeply couple edge prediction with segmentation, we further incorporate edge signal into the above dynamic graph propagation by proposing a double-gate framework:

$$\boldsymbol{h}_{i,t} = \Big(1 - \sigma(\boldsymbol{p}_{i,t}^*) \odot \sigma(\boldsymbol{g}_{i,t})\Big) \odot \boldsymbol{f}_{i,t} + \sigma(\boldsymbol{p}_{i,t}^*) \odot \sigma(\boldsymbol{g}_{i,t}) \odot \boldsymbol{h}_{i,t}^*, \tag{8}$$

where $\boldsymbol{g}_{i,t}$ is defined as:

$$\boldsymbol{g}_{i,t} \triangleq \boldsymbol{W}_e * \boldsymbol{E}[i-1:i+1, t-1:t+1] \tag{9}$$

Note that the edge gating signal is obtained by via a $3 \times 3$ convolution on the $K$-channel edge activation map $\boldsymbol{E}$, which outputs a vector $\boldsymbol{g}$ sharing the same dimension (128) as $\boldsymbol{p}^*$. This way, edges are able to actively influence segmentation via edge-sensitive gating on message passing.

We also hope that the refined segmentation activation after DGP can alternatively serve as a shape regularizer of the edge prediction. To this end, we output a $K$-channel edge regularization feature from $\boldsymbol{F}_m$ using $1 \times 1$ convolution, followed by another $3 \times 3$ convolution to fuse with $\boldsymbol{E}$ to produce a refined edge map $\boldsymbol{E}^*$. This is illustrated in Fig. 1 on the bottom right. One shall see that the above coupled design let the two tasks greatly improve each other, and we term the final framework CSEL.

## 4.4 Training loss

The training loss for CSEL is a multi-task combination of cross-entropy (BCE) losses for multi-label edge learning and the cross-entropy (CE) loss for segmentation:

$$\mathcal{L} = \mathcal{L}_{Seg} + \mathcal{L}_{Edge} = \mathcal{L}_{CE}(\text{Conv}(\boldsymbol{F}^*)) + \lambda(\mathcal{L}_{BCE}(\boldsymbol{E}^*) + \mathcal{L}_{BCE}(\boldsymbol{E})) \tag{10}$$

where $\lambda$ is the parameter controlling the weight of segmentation loss. For segmentation, we use a $3 \times 3$ convolution to linearly project the 128-channel activation $\boldsymbol{F}^*$ into a $K$-channel before the loss.

# 5 Experiments

## 5.1 Datasets and metric

**Cityscapes.** Cityscapes [62] contains 2975 training images, 500 validation images and 1525 private testing images with 19 pre-defined semantic classes. The dataset has been widely adopted as the standard benchmark for both semantic segmentation and semantic edge detection. Following a number of previous works [12, 37, 38, 21], we comprehensively conduct ablation and quantitative studies for both segmentation and edge detection on the validation set.

**SBD.** The Semantic Boundaries Dataset [35] contains both category-level and instance level semantic segmentation annotations. The dataset contains 11355 images (8498 for training and 2857 for testing) from the trainval set of PASCAL VOC2011 [63], and follows the 20-class VOC definition.

**PASCAL VOC 2012** [64] is a semantic segmentation dataset with 1464 (training) and 1449 (val) and 1456 (test) images. We use the augmented dataset with 10582 training images, as in [28]. The dataset contains 20 foreground object classes and 1 background class.

**COCO Panoptic** [10] contains the mask annotations for both things and stuff, with a split setting (118K train 2017 images and 5K validation 2017 images) following the detection community.

**Evaluation metrics.** For evaluation, We consider intersection-over-union (IoU) for segmentation, and the maximum F-score (MF) at optimal dataset scale (ODS) for edge detection[5]. We also consider the boundary F-score proposed in [21] for direct comparison on edge detection task.

## 5.2 Implementation details

**Data loading.** During training, we unify the training crop size as $1024 \times 1024$ for Cityscapes, $472 \times 472$ for SBD and VOC12, and $464 \times 464$ for COCO Panoptic. All models are trained with 150k iterations on Cityscapes with batch size 8, 30k iterations on SBD and VOC12 with batch size 16, and 220k iterations on COCO Panoptic with batch size 16. We also perform data-augmentation with random mirror, scaling (scale factors in $[0.5, 2.0]$) and color jittering.

**Optimization.** we apply an SGD optimizer with a weight decay of $5 \times 10^{-4}$ during training. For baselines and methods that involving CSEL, we additionally apply a second ADAM optimizer to the propagation layers. In our case, we empirically found that ADAM optimizer is more capable of optimizing the propagation layers with better performance.

**Learning rate and loss.** The base learning rates for methods with ResNet-101/ResNet-38 backbones are unified as $3.0 \times 10^{-8}/7.0 \times 10^{-8}$ across Cityscapes, SBD and VOC12. On COCO Panoptic, the base learning rate is unified as $5.0 \times 10^{-8}$ for all comparing methods. Unless indicated, the segmentation weight $\lambda$ in Eq (7) is empirically set it to 0.5 to balance $\mathcal{L}_{Seg}$ and $\mathcal{L}_{Edge}$. We found our method not sensitive to $\lambda$, and setting $\lambda = 0.5$ works excellently for all backbones and datasets.

**Using Mapillary Vistas.** On Cityscapes, a number of literature consider large-scale pretraining on Mapillary Vistas (MV) [65], which is shown to considerably benefit the segmentation performance. Unless indicated, our method does not adopt Mapillary Vistas pre-training for fair comparison.

## 5.3 Experiments on Cityscapes

**Ablation study (semantic segmentation).** Table 1 shows the ablation studies on semantic segmentation where we consider apple-to-apple compared methods that are trained following the same backbones and training protocols: 1) **ST**: Single task segmentation network without propagation layer but keeping the 128-channel feature map $\boldsymbol{F}$. 2) **MT**: The same as ST but naively multi-tasking segmentation with edge detection. 3) **SPN** [13]: Implementation of SPN with the 3-way propagation on $\boldsymbol{F}$, where gating signal is computed using the affinity stream $\boldsymbol{A}$. 4) **SPN+Edge**: Coupling SPN with edge learning following exactly the same double gate design. 5) **DGP**: Baseline which

Table 1: Ablation study of Semantic segmentation on Cityscapes validation set.

| method | ResNet-101 | ResNet-38 |
|---|---|---|
| ST | 77.9 | 78.55 |
| MT | 78.4 | 79.43 |
| SPN | 80.0 | - |
| SPN+Edge | 80.4 | - |
| DGP | 81.3 | - |
| CSEL⁻ | 80.9 | - |
| CSEL | 82.8 | 82.8 |
| CSEL-Ms | **83.7** | **83.4** |

---

[5]we follow [37] by applying exactly the same parameters and settings.

Table 2: Main results of semantic segmentation and semantic edge learning on Cityscapes.

(a) Main semantic segmentation results on the Cityscapes validation set.

| Method | Backbone | MV | mIoU |
|---|---|---|---|
| PSPNet [26] | ResNet-101 | | 78.8 |
| DeepLabV3+ [28] | ResNet-101 | | 78.8 |
| CCNet [66] | ResNet-101 | | 80.5 |
| GSCNN [21] | ResNet-101 | | 74.7 |
| DANet [30] | ResNet-101 | | 81.5 |
| RPCNet [42] | ResNet-101 | | 82.1 |
| CSEL | ResNet-101 | | **83.7** |
| SGPN [57] | ResNet-38 | | 80.9 |
| GSCNN† [21] | ResNet-38 | | 80.8 |
| VRec-JP (LR) [67] | ResNet-38 | | 81.4 |
| Axial-DeepLab [68] | AxiaiRes-XL | | 81.1 |
| CSEL | ResNet-38 | | **83.4** |

(b) Main semantic segmentation results on the Cityscapes test set.

| Method | Backbone | MV | mIoU |
|---|---|---|---|
| PSPNet [26] | ResNet-101 | | 78.4 |
| PSANet [69] | ResNet-101 | | 80.1 |
| ASPP [27] | ResNet-101 | | 80.1 |
| BFP [58] | ResNet-101 | | 81.4 |
| DA-Net [30] | ResNet-101 | | 81.5 |
| OCR [29] | ResNet-101 | | 81.8 |
| CCNet [66] | ResNet-101 | | 81.9 |
| RPCNet [42] | ResNet-101 | | 81.8 |
| CSEL | ResNet-101 | | **82.1** |
| GSCNN | ResNet-38 | ✓ | 82.9 |
| CSEL | ResNet-38 | ✓ | **83.5** |

(c) Main results of semantic edge detection on the Cityscapes validation set. Results measured by Maximum F-Score (MF) at optimal data scale.

| Method | Backbone | MF (IS) | MF (Non-IS) |
|---|---|---|---|
| CASENet [12] | ResNet-101 | 68.1 | 68.9 |
| SEAL [37] | ResNet-101 | 69.1 | - |
| STEAL [38] | ResNet-101 | 69.7 | 71.4 |
| RPCNet [42] | ResNet-101 | - | 78.2 |
| CSEL | ResNet-101 | **78.1** | 78.3 |
| CSEL | ResNet-38 | - | **78.7** |

contains the proposed DGP layer without the double gate design. 6) **CSEL⁻**: Baseline which is the same as CSEL except for removing the edge loss. 7) **CSEL**: Our full method with single scale inference. 8) **CSEL-MS**: CSEL with multiscale inference at $\{0.5, 0.75, 1.0, 1.25, 1.5, 1.75, 2.0\}$. We also consider both instance-sensitive (IS) and non-instance-sensitive (Non-IS) settings where the edge training/evaluation labels are with/without instance-edges.

**Discussions.** We make sure that the comparisons are apples to apples and fair, by using the same backbones and training recipes (such as learning rate, crop size, batch size, and number of iterations) for all comparing methods.The DGP baseline can be considered as an apples to apples counterpart of SPN, whereas the CSEL⁻ baseline is a further study to understand whether the improvement in CSEL purely comes from the enriched representation with edge features. One could observe several trends from the results: 1) The proposed multi-task network consistently outperforms its single task counterpart. 2) CSEL outperforms both SPN and SPN+Edge with the dynamic graph propagation and edge gating. 3) The dynamic graph design in DGP alone helps it to achieve 81.3% mIoU, outperforming both SPN and SPN+Edge by 1.3% and 0.9%, respectively. 4) The incorporation of edge guidance with the double gate design further leads to another non-trivial 1.5% improvement. Simply adding the edge feature does not improve the segmentation quality. In fact, CSEL⁻ is even slightly lower than DGP where no edge features are involved. This reflects the importance of edge signal as a structural guidance than making the representation more expressive.

**Comparison to state-of-the-art (semantic segmentation).** We also compare CSEL to state-of-the art semantic segmentation models and list the results in Table 2a and 2b. For clarity, we divide to table into two parts with models using the same type of backbone putting together. Among the comparing methods, CSEL obtains the best results using both ResNet-101 and ResNet-38 backbones.

**Ablation study (edge detection).** We additionally conduct experiments on edge detection. We first show an ablation study in Table 3, where **ST** indicates single-task edge detection network with the proposed edge stream. **MT** indicates naive multi-task training of both segmentation and edge. **MT** leads to slight performance degradation on the edge detection task compared to single-task

Table 3: Ablation study of edge detection on the Cityscapes validation set. Results measured by MF and scaled by %.

| Method | Res101 (IS) | Res101 (Cls) | Res38 (Cls) |
|---|---|---|---|
| ST | 74.66 | 74.87 | 75.42 |
| MT | 74.48 | 74.62 | 75.38 |
| CSEL | 77.58 | 77.67 | 78.19 |
| CSEL-Ms | **78.09** | **78.32** | **78.73** |

training on edge detection. However, the degradation is marginal compared to the significant improvement of the edge quality introduced by deep coupling of the two tasks in CSEL. Again, this shows the considerable benefit of coupled learning with both segmentation and edge detection.

**Comparison to state-of-the-art (edge detection).** We also compare to state-of-the-art semantic edge detection methods in Table 2c where we present the training/evaluation protocols in two categories (IS and Non-IS) following SEAL. Note that we evaluate the results with the edge thinning protocol. One could see that our method achieves the best performance in both settings

Table 4: Results of boundary evaluation on Cityscapes validation set following [21]. Results measured by boundary F-score and scaled by %. † indicates results reported in [21].

| Method | Backbone | 12px | 9px | 5px | 3px |
|---|---|---|---|---|---|
| DeepLabV3+† | ResNet-38 | 80.1 | 78.7 | 74.7 | 69.7 |
| GSCNN [21] | ResNet-38 | 81.8 | 80.7 | 77.6 | 73.6 |
| CSEL | ResNet-38 | 87.0 | 86.0 | 83.4 | 79.6 |
| CSEL | ResNet-101 | **87.2** | **86.2** | **83.5** | 79.6 |

with significant performance gain from the coupled edge and segmentation learning, as well as the benefit from the dynamic graph propagation module. Besides the evaluation protocol from SEAL, we also follow a separate evaluation protocol from GSCNN [21] in Table 4, where we use the original

evaluation code base from GSCNN to evaluate the boundary quality of semantic segmentation. One can observe that at all different thresholds, CSEL with both ResNet-101 and ResNet-38 backbones outperform DeepLabV3+ and GSCNN by significant margins in terms of boundary quality.

## 5.4 Experiments on SBD

We also evaluate CSEL on the SBD dataset and compare with previous state-of-the-art semantic edge detection models. The main results are presented in Table 5. Note that we used the re-annotated SBD test set from [37] to pursue more precise semantic edge evaluation. From the table, one could see CSEL outperforms other state-of-the-art methods in both IS and Non-IS settings.

Table 5: Semantic edge detection on the re-annotated SBD test set. Results measured by MF and scaled by %. † indicates results are reported by [38].

| Method | Backbone | MF (IS) | MF (Non-IS) |
|---|---|---|---|
| CASENet [12] | ResNet-101 | 63.6 | 63.5† |
| SEAL [37] | ResNet-101 | 67.0 | 66.8† |
| STEAL [38] | ResNet-101 | 65.8 | 68.2 |
| DFF | ResNet-101 | 68.0 | - |
| CSEL | ResNet-101 | **73.1** | **73.2** |

## 5.5 Experiments on PASCAL VOC12

We evaluate CSEL on the PASCAL VOC12 dataset and report the main results of semantic segmentation in Table 6. The performance is evaluated as the mean IoU over the 20 PASCAL VOC classes plus the background class. We compare CSEL with previous competitive methods that are state-of-the-art (DeepLabv3+) or considerably related (SPN/DFN). CSEL achieves considerable improvement these methods.

Table 6: Results on VOC12 validation set.

| Method | Backbone | mIoU |
|---|---|---|
| DT-EdgeNet [52] | ResNet-101 | 69.96 |
| SPN (3-way) [13] | ResNet-101 | 75.28 |
| DFN [41] | ResNet-101 | 80.60 |
| DeepLabv3+ [28] | ResNet-101 | 80.57 |
| CSEL | ResNet-101 | **83.30** |

## 5.6 Experiments on COCO Panoptic

We further conduct experiments on the full COCO Panoptic dataset. Several earlier methods have reported results on COCO-Stuff 10K which is an earlier version of COCO with much less data. However, this does not lead to significant differences with VOC12 in both the size and the diversity of data. We therefore adopt the full COCO Panoptic dataset which is significantly larger and is widely accepted by the detection and instance/panoptic segmentation communities. We hope that our work present solid baselines that inspires subsequent research on this benchmark.

Table 7: Results on COCO Panoptic val2017.

| Method | Backbone | mIoU |
|---|---|---|
| ST | ResNet-101 | 53.95 |
| MT | ResNet-101 | 54.14 |
| SPN | ResNet-101 | 56.49 |
| SPN-Edge | ResNet-101 | 56.51 |
| CSEL | ResNet-101 | **57.03** |

Specifically, we convert the instance segmentation annotations to semantic segmentation ones, and compare segmentation methods with or without the proposed CSEL module. All comparing methods adopt the same CASENet (ResNet-101) architecture. Other hyperparameters are kept exactly the same. Models are trained on the train2017 split and tested on val2017 with single scale inference.

From the results in Table 7, one could see that adding CSEL leads to consistently improved results over the naïve segmentation baselines. In addition, multi-task learning with both segmentation and edge losses also slightly outperforms the single segmentation. Finally, CSEL slightly outperforms SPN with reduced gain compared to results on Cityscapes. We hypothesize that the reduced gain is partly caused by the noisier mask annotations on COCO Panoptic which makes edge learning more challenging and decreases the edge gate quality.

## 5.7 Robustness Against Natural Corruptions

We hypothesize that CSEL also brings robustness to the segmentation model. We show that this is the case and empirically verify on Cityscapes-C in Table 8. Specifially, we follow the standard corruption package provided by [70] to corrupt the Cityscapes validation images. This expands the Cityscapes validation set with 16 types of algorithmically generated corruptions from 4 major categories: "Noise", "Blur", "Weather" and "Digital". Each corruption type also contains 5 severity levels, leading to 2500 evaluation images for each type alone[6].

Our result shows that CSEL overall improves the robustness significantly, especially compared to ResNet-38 and GSCNN [21] which share the same backbone. On the other hand, the method does

---

[6]We only consider level 1-3 for the "Noise" category following [71].

still show some vulnerability to certain type of corruptions, particularly those belonging to the "Noise" category. However, this trend is aligned with ResNet-38 and GSCNN and we hypothesize that it may be partly related to the specific design of ResNet-38 based on this pattern.

Table 8: **Main results on Cityscapes-C.** "DLv3+", "MBv2", "R" and "X" refer to DeepLabv3+, MobileNetv2, ResNet and Xception. The mIoUs of compared methods are reported from [71].

| Arch. | Clean | Blur | | | | Noise | | | | Digital | | | | Weather | | | |
|---|---|---|---|---|---|---|---|---|---|---|---|---|---|---|---|---|---|
| | | Motion | Defoc | Glass | Gauss | Gauss | Impul | Shot | Speck | Bright | Contr | Satur | JPEG | Snow | Spatt | Fog | Frost |
| DLv3+ (MBv2) | 72.0 | 53.5 | 49.0 | 45.3 | 49.1 | 6.4 | 7.0 | 6.6 | 16.6 | 51.7 | 46.7 | 32.4 | 27.2 | 13.7 | 38.9 | 47.4 | 17.3 |
| DLv3+ (R50) | 76.6 | 58.5 | 56.6 | 47.2 | 57.7 | 6.5 | 7.2 | 10.0 | 31.1 | 58.2 | 54.7 | 41.3 | 27.4 | 12.0 | 42.0 | 55.9 | 22.8 |
| DLv3+ (R101) | 77.1 | 59.1 | 56.3 | 47.7 | 57.3 | 13.2 | 13.9 | 16.3 | 36.9 | 59.2 | 54.5 | 41.5 | 37.4 | 11.9 | 47.8 | 55.1 | 22.7 |
| DLv3+ (X41) | 77.8 | 61.6 | 54.9 | 51.0 | 54.7 | **17.0** | **17.3** | 21.6 | **43.7** | 63.6 | 56.9 | 51.7 | 38.5 | 18.2 | 46.6 | 57.6 | 20.6 |
| DLv3+ (X65) | 78.4 | 63.9 | 59.1 | **52.8** | 59.2 | 15.0 | 10.6 | 19.8 | 42.4 | 65.9 | 59.1 | 46.1 | 31.4 | 19.3 | **50.7** | 63.6 | 23.8 |
| DLv3+ (X71) | 78.6 | **64.1** | 60.9 | 52.0 | 60.4 | 14.9 | 10.8 | 19.4 | 41.2 | 68.0 | 58.7 | 47.1 | **40.2** | 18.8 | 50.4 | 64.1 | 20.2 |
| ICNet | 65.9 | 45.8 | 44.6 | 47.4 | 44.7 | 8.4 | 8.4 | 10.6 | 27.9 | 41.0 | 33.1 | 27.5 | 34.0 | 6.3 | 30.5 | 27.3 | 11.0 |
| FCN8s | 66.7 | 42.7 | 31.1 | 37.0 | 34.1 | 6.7 | 5.7 | 7.8 | 24.9 | 53.3 | 39.0 | 36.0 | 21.2 | 11.3 | 31.6 | 37.6 | 19.7 |
| DilatedNet | 68.6 | 44.4 | 36.3 | 32.5 | 38.4 | 15.6 | 14.0 | 18.4 | 32.7 | 52.7 | 32.6 | 38.1 | 29.1 | 12.5 | 32.3 | 34.7 | 19.2 |
| ResNet-38 | 77.5 | 54.6 | 45.1 | 43.3 | 47.2 | 13.7 | 16.0 | 18.2 | 38.3 | 60.0 | 50.6 | 46.9 | 14.7 | 13.5 | 45.9 | 52.9 | 22.2 |
| PSPNet | 78.8 | 59.8 | 53.2 | 44.4 | 53.9 | 11.0 | 15.4 | 15.4 | 34.2 | 60.4 | 51.8 | 30.6 | 21.4 | 8.4 | 42.7 | 34.4 | 16.2 |
| GSCNN | 80.9 | 58.9 | 58.4 | 41.9 | 60.1 | 5.5 | 2.6 | 6.8 | 24.7 | 75.9 | 61.9 | 70.7 | 12.0 | 12.4 | 47.3 | 67.9 | 32.6 |
| CSEL (R38) | **82.8** | 63.7 | **66.4** | 44.3 | **66.4** | 15.1 | 6.4 | 17.5 | 40.4 | **79.7** | **66.0** | **78.3** | 17.1 | **19.5** | 41.4 | **78.5** | **33.0** |

# 6    Conclusion

We proposed a unified framework for coupled segmention and edge learning. Our work revisits the two long-standing and important perceptual grouping problems - semantic segmentation and edge detection. Our method (CSEL) includes a novel end-to-end multi-task network, a recurrent dynamic graph propagation layer, as well as deep coupling of the two tasks on top on top of them. Finally, results show that careful coupling of the tasks leads to significant improvement on both of them.

## Broader Impact

Our method coupled two long-standing important vision problems, semantic segmentation and edge detection, under a unified framework. Besides the various practical benefits from deeply coupling these two tasks, we expect the research to inspire considerable insights, interests and revisits on perceptual grouping, mid-level representations and structured prediction. The result of the research is likely to find diverse scene understanding applications such as autonomous driving and robot navigation. Like many other discriminative recognition models, our method inevitably faces challenges from input data quality and underlying data biases. The model behavior is subject to various factors such as data distributions, domain gaps, label quality and fairness. We encourage researchers to focus on the "in the wild" robustness when the above challenges are present.

## Acknowledgement

We thank the NVIDIA GPU Cloud (NGC) team for the computing support of this work. We also thank the anonymous reviewers and the other NVIDIA colleagues who helped to improve this work with discussions and constructive suggestions.

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
