# Supplementary: Coupled Segmentation and Edge Learning via Dynamic Graph Propagation

**Zhiding Yu**[*]  **Rui Huang**[*†],  **Wonmin Byeon,  Sifei Liu,  Guilin Liu,**
**Thomas Breuel,  Anima Anandkumar,  Jan Kautz**

NVIDIA

## A    Additional implementation details

Due to space limit in the main paper, we describe some additional implementation details here.

**Mapillary Vistas pre-training.** Our experiment with ResNet-38 on the Cityscapes Test set involves pre-training on Mapillary Vistas. In particular, we adopt the same data loading protocol (random mirror, scaling and color jittering), crop size ($1024 \times 1024$) and base learning rate ($3 \times 10^{-7}$) as training on Cityscapes. Since Mapillary Vistas in general contains larger images than Cityscapes, we adjust the scale factor to be within $[0.5, 1.5]$ instead of $[0.5, 2.0]$. The total number of training iterations is set to 500K.

**Training pipeline for Cityscapes test.** For Cityscapes test evaluation, we include the validation set data in model training following previous works. We first train 150K iterations on the Cityscapes training set, using the validation set to validate the hyperparameters and pick the checkpoint. We then start from this model and fine-tune another 50K iterations on the combined trainval set. The final model is used for test set evaluation.

All our models (ResNet-101/38) are trained on the Cityscapes data with fine annotations. We do not include any coarsely labeled Cityscapes data during the training process.

**Learning rate policy.** For all our experiments, we adopt the "poly" learning rate decay policy with power $0.9$. We apply sum reduction where the final segmentation/edge losses are computed by summing up the losses across pixels. To avoid possible gradient explosion and stabalize training, we apply a quadratic warmup policy with $5K/1K$ iterations before the main training loop.

**Computation resources.** All experiments are conducted on a single node with 8 NVIDIA Tesla V100 GPUS (32G). For Cityscapes experiments with batch size 8, iteration number 150K, and crop size $1024 \times 1024$, CSEL training takes less than 6 days on a ResNet-101 backbone and less than 7 days on a ResNet-38 backbone. For SBD and VOC12 experiments with batch size 16, iteration number 30K, and crop size $472 \times 472$, CSEL training takes about 12 hours on a ResNet-101 backbone. Note that we have not adopted acceleration techniques such as mixed-precision training with APEX.

## B    Additional visualizations

**Additional visualization on Cityscapes.** We present qualitative results of CSEL in Figure 1 and its comparison with the multi-task baseline model. Our method produce sharper edges and smoother semantic segmentation, especially in cluttered area. In terms of semantic segmentation, our method generally give more accurate prediction near object or semantic boundaries, and has less irregular shape or hole in its prediction.

---

[*]Equal contribution. Correspondence to Zhiding Yu <zhidingy@nvidia.com>.

[†]Work partially done during an internship at NVIDIA.

35th Conference on Neural Information Processing Systems (NeurIPS 2021).

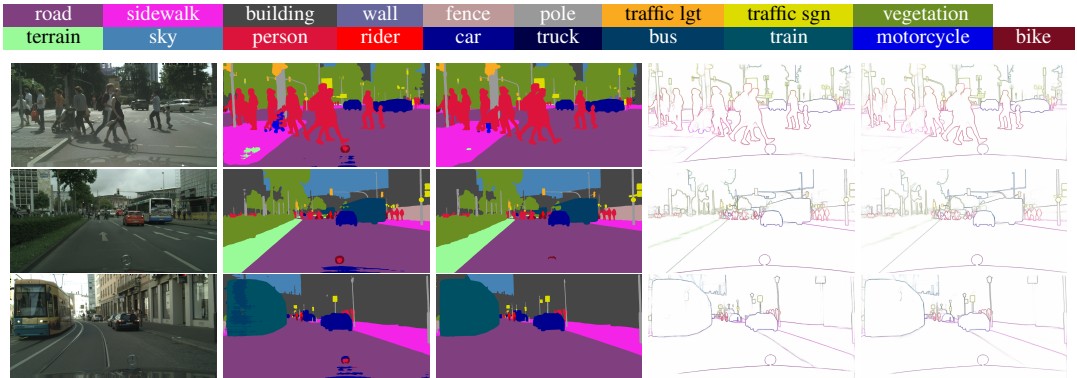

Figure 1: Qualitative results on Cityscape validation set. From left to right: original image, segmentation of multi-task backbone (Res38) and CSEL-Res38, semantic edge prediction of multi-task backbone (Res38) and CSEL-Res38. Best viewed in color and zoom in.

we further visualize more semantic segmentation and semantic edge detection results of CSEL ResNet-101 (IS) and CSEL ResNet-38 (Non-IS) in Figure 2. Note that one could see a major difference on the instance-sensitive edges of two neighboring objects sharing the same category.

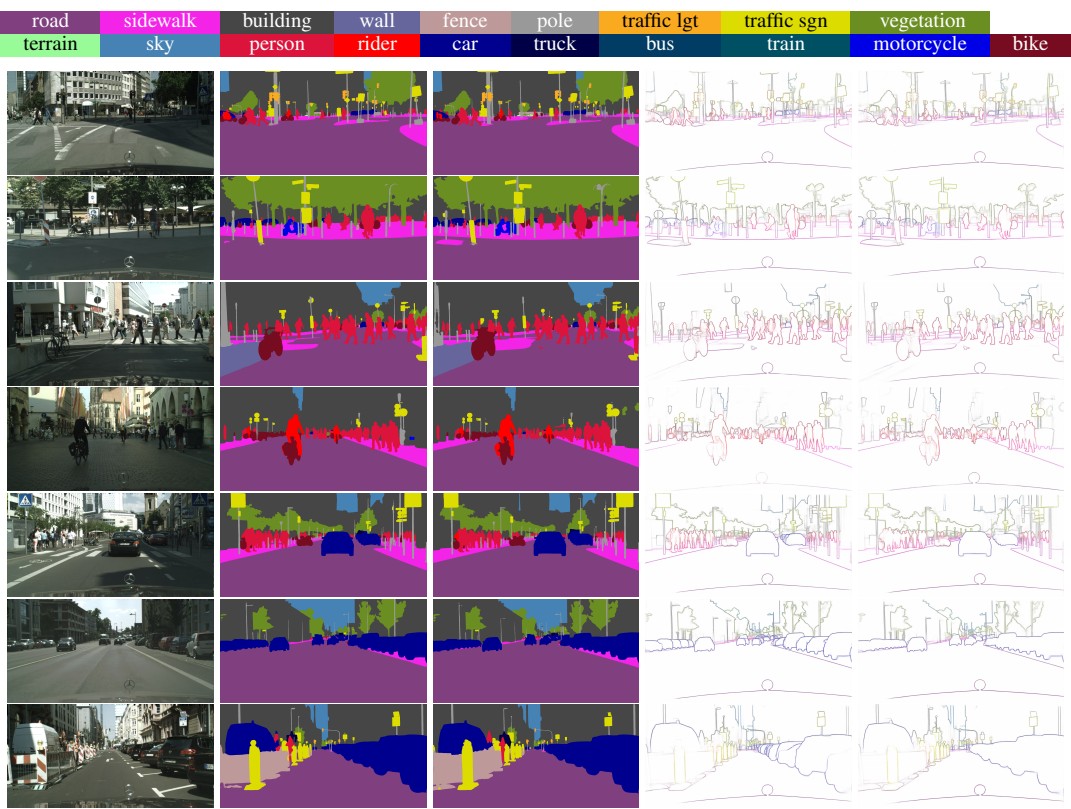

Figure 2: Qualitative results on Cityscape validation set. From left to right: original image, segmentation of CSEL-Res101 (IS) and CSEL-Res38 (Non-IS), semantic edge prediction of CSEL-Res101 (IS) and CSEL-Res38 (Non-IS). Best viewed in color and zoom in.

**Additional visualization on SBD** We visualize the semantic edge detection results on SBD and illustrate them in Figure 3. One could see that the proposed method is able to produce high quality sharp edge predictions without using any edge alignment techniques in [1] and [2].

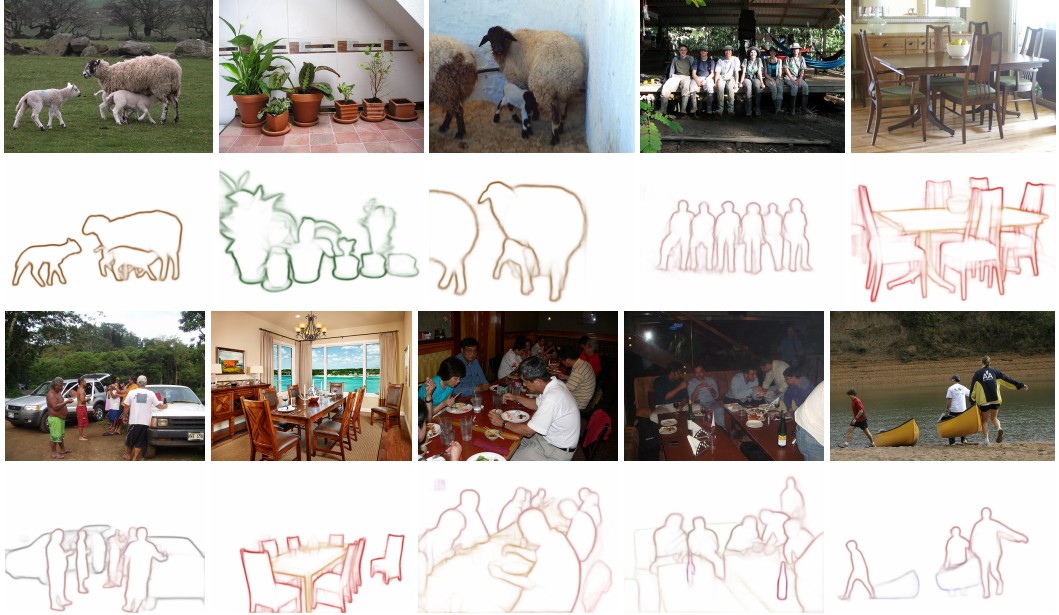

Figure 3: Qualitative results on SBD re-annotated test set. Best viewed in color and zoom in.

**Additional visualization on VOC12.** We visualize the semantic segmentation results on PASCAL VOC12 and illustrate them in Figure 4. One could see that the proposed method is able to produce spatially smoothed predictions while capturing structural details.

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

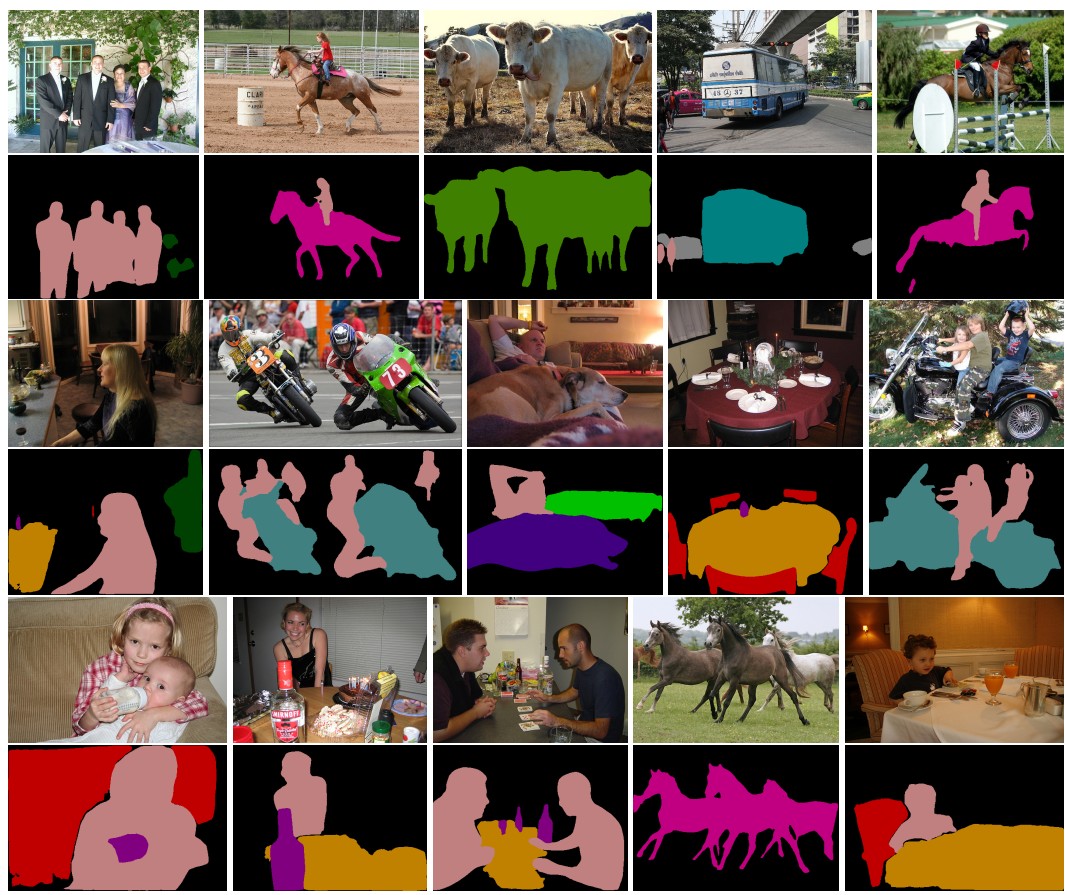

Figure 4: Qualitative results on PASCAL VOC 12 val set. Best viewed in color and zoom in.