# OpenReview forum: "Coupled Segmentation and Edge Learning via Dynamic Graph Propagation"
_NeurIPS.cc/2021/Conference — NeurIPS 2021 Poster_

### Official Review · Reviewer_vms1 · 2021-07-15

**Rating:** 6
**Confidence:** 3

**Summary:**

The paper proposes coupling edge prediction and semantic segmentation using a multi-task network dubbed CSEL. Some of their goals include the mutual improvement of performance on the two tasks when learned together. The proposed model (CSEL) extends spatial propagation network SPN  with a dynamic graph propagation (DGP) scheme. DGP is a gated recurrent graph message passing that is responsible for selecting neighboring pixels with the highest response. In DPG, the semantic edges are considered as gating signals to refine semantic segmentation.

**Ethical Concerns:**

No.

**Limitations And Societal Impact:**

Yes.

**Main Review:**

The proposed method is nontrivial and provides some additional gains over SOAT. The paper also contains an insightful ablation study, which is positive. In many experiments, the performance gains are not very expressive, which makes me feel lukewarm about this approach. A major issue to me seems to be the lack of clarity in the explanations and figures. I feel it is hard for a reader to understand the concepts/architecture and, replicate the work for future research. I would not recommend this paper for publication in this current state, however, I will keep an open mind for post rebuttal remarks. Some more detailed comments are presented below:

1) I would appreciate hearing from the authors on why the edges only serve segmentation only as gating signals?  Could it not be helpful to enrich the segmentation features as well? Actually, I believe this is one point where clarification could be improved. The message passing figures seem a bit confusing to me and the legends are not informative.

2) Figure 1, again the legend could be improved. There are elements in the figure that are not explained in the legend, which forces the reader to look for them in the text. In the same figure, it does not show how the message passing is happening in DGP and which features are involved, which makes it harder to understand the overall framework.

3) Clearer comparison with SPN, since the model seems to build over SPN, I was expecting a closer analysis on the difference between them plus, the exact difference in terms of technical components.

4) I was expecting some qualitative results of CSEL in the mains paper (there are some in suppl. which is ok). However, I could not find any qualitative comparison of CSEL and SPN, and I was curious about how the performance gains of CSEL translate visually against SPN+edge, for example.

**Time Spent Reviewing:**

4

---

> ### Author Response · Authors · 2021-08-10
> **Author Response to Reviewer 3 (vms1)**
>
> **Q1: More expressive breakdown of the performance gain with the DGP baseline**
>
> A1: We thank the reviewer for pointing out this weakness. We understand the reviewer’s concern and will add an additional analysis subsection in the experiments section to specifically cover all the subsequent discussions, including new baselines, the role of edge features, clearer comparison with SPN, and qualitative comparison.
>
> To render a more expressive analysis of the performance gain and to lay down a foundation for subsequent discussions, we first conduct two additional single segmentation baselines:
> 1) A DGP baseline which only contains the proposed DGP layer by removing both the edge branch and losses in CSEL.
> 2) A CSEL- baseline which keeps everything the same as CSEL but only removes the edge loss.
>
> For both baselines in particular, we adopt the CASENet (ResNet-101) backbone architecture and keep all the other hyperparameters such as learning rate, crop size, batch size, and number of iterations the same as reported in the main paper. We train the network on Cityscapes training set, and test on the validation set following the other methods in Table 1 of the main paper.
>
> The DGP baseline can be considered as an apples to apples counterpart of SPN, whereas the CSEL- baseline is a further study to understand whether the improvement in CSEL purely comes from the enriched representation with edge features. The baselines give a breakdown of the performance gain in CSEL:
> 1) The dynamic graph design in DGP alone helps it to achieve 81.3% mIoU, outperforming both SPN and SPN+Edge by 1.3% and 0.9%, respectively.
> 2) The incorporation of edge guidance with the double gate design further leads to another non-trivial 1.5% improvement.
> 3) Simply adding the edge feature does *not* improve the segmentation quality. In fact, the performance of CSEL- is even slightly lower than DGP where no edge features are involved. This also reflects the importance of edge signal as a guidance, than purely an enrichment to the representation.
>
> The results are listed in the following table for comparison. All results are obtained with single scale inference.
>
> | Method       	| Single task    | Multi-task	 |  SPN	        | SPN+Edge    |    DGP           |      CSEL-	|  CSEL      |
> |:------------------:|:----------------:	|:-------------------:|:--------------:	|:-----------------:	|:-----------------:	|:-----------------:	|:-----------------:	|
> | mIoU (%) 	|       77.9         |       78.4     	  |      80.0        |      80.4 	        |   81.3    	        |       80.9         |      82.8         |
>
>
> The result does show the non-negligible positive impact from edge learning. The next question would be why and how edge signals helped to obtain such improvement, as we will answer subsequently. We will also improve the clarity in the explanations and figures.
>
> **Q2: Discussion on the role of edge features**
>
> A2: There are many ways a feature can get “enriched”. The proposed CSEL module itself can also be broadly considered as a structured enrichment of the segmentation feature, where edge features provide an alternative view of local contrast details that are originally not well captured, and influence the segmentation in terms of gating signal in message pasing. By saying “enrich”, we assume that the reviewer means making the model “wider and more expressive”, such as concatenating and fusing the edge features with segmentation features.
>
> Naively enriching the feature through concatenation and fusion does not lead to significant improvement. The idea is straightforward and we have in fact tried similar ones before, but have seen limited improvements in segmentation quality and sometimes even negative results. A fundamental reason is that edges are very local and thin structures. Naively fusing it with segmentation through limited layers of convolutions has very little chance to influence segmentation predictions through long-range interactions. This is why message passing and gating should be introduced.
>
> Another related idea is to use eigendecomposition based globalization to transform edge maps into eigen maps which contain solid regions of foreground/background segmentations before feeding into a network [a]. We recommend the reviewer read [a] and several other pioneer works such as [b] for more information. In any case, edge maps cannot be directly used to fuse with segmentation without proper transforms.
>
> [a] Bertasius et al., High-for-low and low-for-high: Efficient boundary detection from deep object features and its applications to high-level vision, ICCV 2015.
> [b] Arbelaez et al., Contour detection and hierarchical image segmentation, T-PAMI 2010.
>
> **Q3: Clearer comparison with SPN**
>
> A3: The core differences between SPN and CSEL in terms of technical components are:
>
> 1) SPN adopts a three-way connection design as shown in Fig. 3 of the main paper, while the DGP contains a dynamic one-way connection design as mathematically described in Equation (5) - (7) and Figure 4 of the main paper. This design allows dynamically constructed graphs that sparsify the message passing routes and better capture the inter-pixel similarity relationship. The design thus leads to improved performance over SPN, as shown in the table above.
>
> 2) CSEL contains a double gate design which does not exist in SPN, as mathematically described in Equation (8) - (9) of the main paper. This design allows the proposed DGP layer to also incorporate edge cues as gating signals to block the messages near boundaries.
>
> 3) A final key difference is that the one-way connection design in DGP helps CSEL to avoid the complicated normalization as a result of the three-way connection in SPN. The simplification leads to improved learning behavior, especially when edge signals are incorporated through the double gate design. This is verified by the results in the above table, where the gain of CSEL over DGP is more significant than SPN+Edge over SPN. The gain is also more significant than that of naive multi-tasking, even though with a higher level base performance. The result shows that DGP is a well-motivated design as it magnifies the performance gain from edge learning.
>
> Other than the above differences, we have made sure that the comparisons are apples to apples and fair, by using the same backbones, gate input and training recipes for all comparing methods.
>
> **Q4: Qualitative comparison between CSEL and SPN/SPN+Edge**
>
> A4: On most regular images, the two methods are visually quite similar. SPN is also a strong deep structured model which smooth the segmentation feature through message passing. As a result, they behave similarly, such as completing the disconnected thin structures (such as pole in Cityscapes), filling wrong predictions/holes and removing aliasing effects on very large objects. Slight differences, however, do emerge when they handle more difficult categories such as wall and fence. SPN tends to show slightly more failure cases where structured prediction cannot entirely save the bad unary prediction. This is probably caused by the more complicated normalization in SPN. Similar failure cases happen to CSEL as well. We will add more visual examples to the final paper.

---

> > ### Comment · Reviewer_vms1 · 2021-09-01
> > **Final thoughts**
> >
> > I would like to thank the authors for the detailed explanations. After reading all the other reviews and the rebuttal, I am a bit more positive about this paper. The authors addressed most of my concerns regarding their method. However, I still believe the paper could improve in terms of presentation as mentioned in my main review.

---

> > > ### Author Response · Authors · 2021-09-01
> > > **Thank You**
> > >
> > > We thank the reviewer again for the kind support of this work. Your constructive feedback and criticisms will help us greatly towards improving this work.

---

### Official Review · Reviewer_YztE · 2021-07-17

**Rating:** 6
**Confidence:** 5

**Summary:**

This paper proposed a unified framework for detecting edges and segmenting objects. They conduct experiments on Cityscapes, SBD and Pascal VOC. The edge detection results are competitive to previous SOTA methods.

**Limitations And Societal Impact:**

Yes

**Main Review:**

This paper addresses the problem of joint predicting the segmentation results and the edge. By the proposed DGP layer, they achieve good results among several benchmarks. They also show robustness by experimenting on Cityscapes-C. There are some minor concerns, which need more clarification:
1. Do the PASCAL VOC and SBD train together? Are the method in Table 5 and Table 6 also trained under multi-task settings?

2. I am wondering about the efficiency of the proposed DGP. It will be better if the computational cost and inference speed could be compared and discussed.

3. Will edge detection help the semantic segmentation? It will be better if the single edge detection and single segmentation results could be reported in Table 2.

4. There are some typos, for example, Line 78...

**Time Spent Reviewing:**

6

---

> ### Author Response · Authors · 2021-08-10
> **Author Response to Reviewer 2 (YztE)**
>
> **Q1: Clarification on VOC/SBD training**
>
> A1: VOC12 and SBD are trained separately because they have different training, validation (and testing) splits, even though most of their images are shared. The VOC12 mentioned in our work and typically followed by many semantic segmentation works (such as DeepLabv2/DeepLabv3+/SPN) is an augmented one with additional segmentation annotations from the SBD dataset. The very original VOC12 dataset does not contain so many semantic segmentation labels as of today. Despite the strong connections between VOC12 and SBD, their data splits remain different for historical reasons. We separate the training and follow the protocol/split of each one for apples to apples comparisons with previous works.
>
> Yes, for the CSEL methods reported in Table 5 and 6, they are also trained under the multi-task settings.
>
> **Q2: Computational cost and inference speed**
>
> A2: We benchmarked the inference speed with and without the CSEL module following the suggestion. In particular, we benchmark with the CASENet (ResNet-101) architecture on an NVIDIA Tesla V100 GPU. With an input size of 1024 x 2048, the naive multi-tasking network shows an average inference time of 0.98s/image, whereas CSEL further adds a moderate 0.57s overhead to the inference time (1.55s/image in total).
>
> Note that, the CSEL module is currently based on a PyTorch level implementation in which the inference speed has space to be further optimized. Specifically, the propagation is done via a Python "for" loop with purely PyTorch operations (packaged inside a custom subclass of RNNCellBase), rather than using CUDA to speed up the iteration over a sequence of 128 length. Complexity-wise, the proposed module does not introduce much overhead since all the graph connections are restricted to the immediate neighbors. Therefore the complexities of the DGP and CSEL layers are at the same level as regular 3x3 convolutions. The true bottleneck that’s actually causing most of the overhead in the real inference speed is the sequential recurrent operations.
>
> **Q3: Whether edge helps segmentation, and single edge detection/segmentation results**
>
> A3: We have shown that under *naive* multi-tasking settings, edge detection slightly helps semantic segmentation compared to single semantic segmentation, while segmentation causes the edge detection performance to be slightly degraded compared to single edge detection. In particular, Table 1 and Table 3 show the segmentation and edge detection results on Cityscapes validation, respectively. In Table 1, ST indicates the naive single segmentation baseline, while MT indicates the proposed naive multi-task architecture. Similar for Table 3, where ST indicates the naive single edge detection baseline, while MT shares the same model with the MT in Table 1.
>
> Per request from R3, we additionally report two single segmentation baselines:
> 1) DGP which is applying the pure dynamic graph layer without any edge branch features and guidance.
> 2) CSEL- which is exactly the same as CSEL, except that removing the edge loss (but still keeping the edge features).
>
> The improved segmentation performance of CSEL over both DGP and CSEL- again shows the non-negligible positive impact of edge learning as a structured guidance in segmentation tasks. Please kindly refer to our response to R3 for more details.
>
>
> **Q4: Typos**
>
> A4: Many thanks. We will correct these typos in future versions.

---

> > ### Comment · Reviewer_YztE · 2021-09-02
> > **Final decision**
> >
> > The author has addressed my concerns. It is more clear with the new provided experiment results. Toward the newly added experiments, I vote for acceptance.

---

> > > ### Author Response · Authors · 2021-09-02
> > > **Thank You**
> > >
> > > Thank you for the kind support and constructive feedback!

---

### Official Review · Reviewer_qLf1 · 2021-07-20

**Rating:** 7
**Confidence:** 4

**Summary:**

This paper proposes a dynamic (i.e. recurrent) graph propagation module to improve the results of a feedforward semantic segmentation algorithm on the Cityscapes dataset. In addition to outputting the typical feature map where each pixel is assigned to one of 19 categories (i.e. the semantic segmentation map), the proposed network produces supervised predictions of semantic _edges_ (the boundaries between classes) and a set of local pixel-pixel affinities. These two novel outputs are used as gates in a recurrent message-passing algorithm, which refines the semantic segmentation map (as well as the edge map.) The authors show that the refined map gives improved semantic segmentation scores over the unadorned baseline and a number of other state-of-the-art algorithms. Finally, they find that the dynamic graph propagation module gives substantial improvements on "corrupted" versions of the Cityscape dataset (at least for some types of corruption.)

**Ethics Review Area:**

["I don’t know"]

**Limitations And Societal Impact:**

Yes

**Main Review:**

The authors' approach is intuitive and clearly explained, as is the logic for why information about edges should be useful for recurrently refining a feedforward feature map. They thoroughly test their model and ablations on Cityscapes, and I buy that adding their module yields an improvement.

The performance improvements on some types of image corruptions are especially interesting. I wish the authors had included a bunch of qualitative examples and tried to analyze why their module is helpful in dealing with some types of corruption but not others. Indeed, the results in this section are strong enough that it's arguably their most important contribution, since the Cityscapes semantic segmentation task is so close to performance ceiling anyway (and adding a few percentage points to mIoU may not mean detecting new objects that were being missed before, etc.)

Other than this, my main suggestion is that the authors evaluate their model on at least one, and ideally several other datasets and tasks. Cityscapes semantic segmentation is peculiar in that there are not too many categories and the images are pretty homogeneous. If their module significantly improved results on a more diverse dataset (e.g. COCO) I'd be more inclined to think that they had really identified a very useful novel computation for the segmentation task. Likewise, it seems that their approach could be adopted for performing or improving instance segmentation (rather than semantic segmentation), which is a task in much greater need of algorithmic improvement than the latter. If the authors could offer more than this one example (Cityscapes semantic segmentation) of how Dynamic Graph Propagation with joint edge learning improves results, it would much more strongly support their initial motivation for introducing this architecture.

---- Updated ----

**Time Spent Reviewing:**

2.5

---

> ### Author Response · Authors · 2021-08-10
> **Author Response to Reviewer 1 (qLf1)**
>
> **Q1: Additional analysis on Cityscapes-C**
>
> A1: We notice that the proposed CSEL module helps the model to better grab the image gist and context with the spatial propagation. This really helped the model to overcome many corruptions from the Blur and Digital categories.
>
> We hoped that the proposed module could help to overcome Noise type corruptions and JPEG/Snow, because intuitively message passing should help to promote smoother and more compositional feature representations. It turns out, however, that the CSEL module is currently designed to stay at the backend of the network, while such noises tend to have great impacts on the backbone. As a result, the unary predictions from the backbones are often too bad for the proposed module to fix.
>
> We thank the reviewer for this question and will add more detailed analyses/visualizations in future versions.
>
> **Q2: Semantic Segmentation on the COCO Panoptic dataset**
>
> A2: Thanks for the constructive suggestion. In fact, we did conduct some experiments on the full COCO Panoptic dataset, but had the concern that fewer works have systematically reported on this one. On the other hand, several earlier methods have reported results on COCO-Stuff 10K which is an earlier version of COCO with much less data. It does not lead to significant differences with VOC12 in both the size and the diversity of data, as the reviewer is probably looking for. We adopted the full COCO Panoptic dataset since it contains the latest annotations and settings (118K train 2017 images and 5K validation 2017 images) widely accepted by the detection and instance/panoptic segmentation communities. We agree with the reviewer that experiments on COCO would make our method more convincing.
>
> Specifically, we convert the instance segmentation annotations to semantic segmentation ones, and compare segmentation methods with or without the proposed CSEL module. We adopt the same CASENet (ResNet-101) architecture and other network designs as reported in the main paper. For all comparing methods, we unify the learning rate as 5.0 x 10^-8, batch size as 16, random crop size as 464 x 464. Other hyperparameters are kept exactly the same as reported in the paper and supplementary. Models are trained on the train 2017 images and tested on the validation 2017 images.
>
> From the results, one could see that adding CSEL leads to significantly improved results over the naïve segmentation baselines. In addition, multi-task learning with both segmentation and edge losses also slightly outperforms the single segmentation.
>
> | Method       	| Single task    | Multi-task	 |  SPN+Edge	|      CSEL	|  CSEL-Ms      |
> |:------------------:|:----------------:	|:-------------------:|:--------------:	|:-----------------:	|:-----------------:	|
> | mIoU (%) 	|       53.95       |       54.14     	  |      56.51      |      57.03 	|    58.39    	|

---

> > ### Comment · Reviewer_qLf1 · 2021-09-02
> > **Good addition**
> >
> > Thank you for your additional experiment on COCO. It seems that your proposed architecture also leads to improvement on a more general and more challenging semantic segmentation task/dataset, which was one of my main concerns. I have raised my review score and recommend acceptance.

---

> > > ### Author Response · Authors · 2021-09-02
> > > **Thank You**
> > >
> > > Thank you for the kind support and constructive feedback!

---

### Author Response · Authors · 2021-08-10
**Author Response to All Reviewers**

We would like to thank the reviewers for the constructive feedback and for sharing the significance of this work. Below, we respond to each reviewer separately and address all the questions. In summary, we present the following additional results and discussions per request from each reviewer:

1) Additional analysis on Cityscapes-C. (R1)
2) Semantic segmentation on the COCO Panoptic dataset. (R1)

3) Clarification on VOC/SBD training. (R2)
4) Computational cost and inference speed. (R2)
5) Clarification on whether edge helps segmentation, and single edge detection/segmentation results. (R2)

6) More expressive breakdown of the performance gain with the DGP baseline (R3)
7) Discussion on the role of edge features. (R3)
8) Clearer comparison with SPN. (R3)
9) Qualitative comparison between CSEL and SPN+Edge. (R3)

Other concerns on typos, clarity and figures will be carefully addressed.

---

### Decision · Program_Chairs · 2021-09-27

**Decision:**

Accept (Poster)

**Comment:**

The reviewers appreciated the clear and intuitive presentation of the proposed graph propagation strategy. The author responses addressed well the requests for clarification and additional experiments made by reviewers, including results on COCO, the discussion of computational cost, and the details of performance gains over DGP. All reviewers recommend acceptance. The ACs concur with this recommendation.